# RTS Smoother-Guided Learning of Physics-Based Neural Differential Models

## Abstract

Ordinary differential equations (ODEs) are widely used to model dynamical systems in physics, biology, neuroscience, and physiology, but in many applications some equations of the dynamics are unknown and only a subset of the state variables are measured. We propose a hybrid neural–physics framework in which the known components of the ODE are kept explicit and the missing components are represented by a neural network. The proposed method consists of two stages where we alternate between state and parameter estimation and iterate until a predetermined criterion is met. Specifically, in the first step, we treat the model parameters as being known and we infer the latent states from the available measurements using a Rauch–Tung–Striebel (RTS) smoother. In the second stage, we treat the smoothed trajectories as being known and use them to estimate the neural networks' parameters through backpropagation. We evaluate the method on benchmark systems spanning linear, nonlinear, and stiff dynamics under partial state observation. Across these settings, the proposed method learns missing ODE components from incomplete measurements while exploiting and retaining interpretable mechanistic structure and improving latent-state reconstruction and long-horizon prediction.

## 1 Introduction

Ordinary Differential Equations (ODEs) are a standard tool for describing how dynamical systems evolve over time in physics, biology, neuroscience, medicine, and engineering. In practice, however, the equations are often not fully known and the full system state is rarely measured. Instead, we would like to use available measurements to recover hidden states and learn the missing parts of the dynamics. This is challenging for two main reasons:

- **Unknown or incomplete dynamics.** Mechanistic models are often only partially specified: important terms may be missing, feedback mechanisms may be simplified, and subject-specific effects may not be captured by a single closed-form ODE.

- **Partial and noisy measurements.** Typically, only a subset of the state variables is observed, often indirectly and with noise (e.g., membrane voltage but not gating variables, or aggregate flow rather than local pressures), making full-state reconstruction difficult.

Hybrid approaches that combine mechanistic models with learned components have been increasingly used to model dynamical systems with incomplete physics or missing terms (Rackauckas et al., 2020; Imbiriba et al., 2022; Demirkaya et al., 2024). In these approaches, neural networks are used to represent the unknown parts of an ODE, while the known mechanistic terms remain explicit. In many existing neural-ODE and hybrid formulations, however, latent states are inferred using neural encoders or recurrent architectures (Rubanova et al., 2019; De Brouwer et al., 2019). Under strong noise, irregular sampling, or severe partial state

---

[1]The implementation of the proposed method and utilities for adding new ODE systems are available at `https://github.com/PLACEHOLDER/PLACEHOLDER`.

measurement, these learned inference mechanisms can be brittle and their latent representations difficult to interpret.

We take a different route and couple a physics-based neural differential model with classical state-space estimation. Concretely, we treat the known parts of the dynamics as fixed ODE terms and use a neural network to represent the *unknown* differential equations of the system. This yields a hybrid model that combines mechanistic structure with neural networks that approximate the unknown differential equations. The main challenge is to infer both the state and the neural networks' parameters using a set of measurements. To address this challenge, in this paper we propose an iterative two-stage algorithm for state and parameter estimation (see fig. 1). In the first stage, we fix the neural network parameters and use Rauch–Tung–Striebel (RTS) smoothing to estimate the state trajectories and their uncertainties from a set of measurements. In the second stage, we treat the smoothed trajectories as if they were the ground truth and estimate the neural network parameters using backpropagation. We iterate between these two stages until we meet a predetermined criterion. At the end of this process the hybrid model can simulate the dynamics of the system allowing us to creating a digital twin of the real system.

The proposed framework enables researchers to (i) determine all the state variables from noisy partial measurements while respecting the known dynamics; (ii) learn the missing ODEs; and (iii) retain interpretability by preserving the known part of the model.

## 1.1 Problem setting

We consider partially observed trajectories generated by an underlying continuous-time dynamical system with latent state $\boldsymbol{x}(t)$ and measurements $\boldsymbol{y}(t_k)$. The dynamics are modeled as

$$\dot{\boldsymbol{x}}(t) = \boldsymbol{f}_\theta(\boldsymbol{x}(t)) + \boldsymbol{w}(t), \qquad \boldsymbol{y}(t_k) = \boldsymbol{h}(\boldsymbol{x}(t_k)) + \boldsymbol{\varepsilon}_k, \tag{1}$$

where $\boldsymbol{f}_\theta$ is a (partially known) dynamics model with parameters $\theta$, $\boldsymbol{h}$ maps latent states to measurements, and $\boldsymbol{w}(t)$ and $\boldsymbol{\varepsilon}_k$ denote process and measurement noise, respectively. We observe $N$ noisy sequences

$$\mathcal{D} = \big\{\boldsymbol{y}_{1:T_i}^{(i)}\big\}_{i=1}^N, \qquad \boldsymbol{y}_{1:T_i}^{(i)} = \big(\boldsymbol{y}_1^{(i)}, \ldots, \boldsymbol{y}_{T_i}^{(i)}\big),$$

while the corresponding latent trajectories $\boldsymbol{x}_{0:T_i}^{(i)}$ remain unobserved. The discrete-time state-space model induced by sampling eq. (1) is given in section 3. We would like to find $\theta$ such that rollouts of the model, started from appropriate latent states, produce predictions that match the measurements while remaining consistent with the underlying latent evolution. Assuming that the $N$ sequences are independent and share a common parameter vector $\theta$, we write the learning problem as minimization of the negative joint log-likelihood:

$$\theta^\star = \arg\min_\theta \left( -\sum_{i=1}^N \log p\big(\boldsymbol{x}_{0:T_i}^{(i)}, \boldsymbol{y}_{1:T_i}^{(i)}\big) \right). \tag{2}$$

Using the product rule, the joint density for each sequence can be decomposed into a measurement term and a latent-state term:

$$\theta^\star = \arg\min_\theta \left( -\sum_{i=1}^N \log \left[ p\big(\boldsymbol{y}_{1:T_i}^{(i)} \mid \boldsymbol{x}_{0:T_i}^{(i)}\big) \, p\big(\boldsymbol{x}_{0:T_i}^{(i)}\big) \right] \right)$$

$$= \arg\min_\theta \left( -\sum_{i=1}^N \log p\big(\boldsymbol{y}_{1:T_i}^{(i)} \mid \boldsymbol{x}_{0:T_i}^{(i)}\big) - \sum_{i=1}^N \log p\big(\boldsymbol{x}_{0:T_i}^{(i)}\big) \right). \tag{3}$$

This makes explicit that the objective contains two contributions: a *measurement-model* term, which encourages predicted observations to match the data, and a *state-model* term, which encourages the latent trajectory to remain consistent with the dynamics.

We now expand each of these two terms using the structure of the state-space model. Since $\boldsymbol{y}_k$ depends only on $\boldsymbol{x}_k$ through $\boldsymbol{h}$, the measurements are conditionally independent given the states, so the measurement

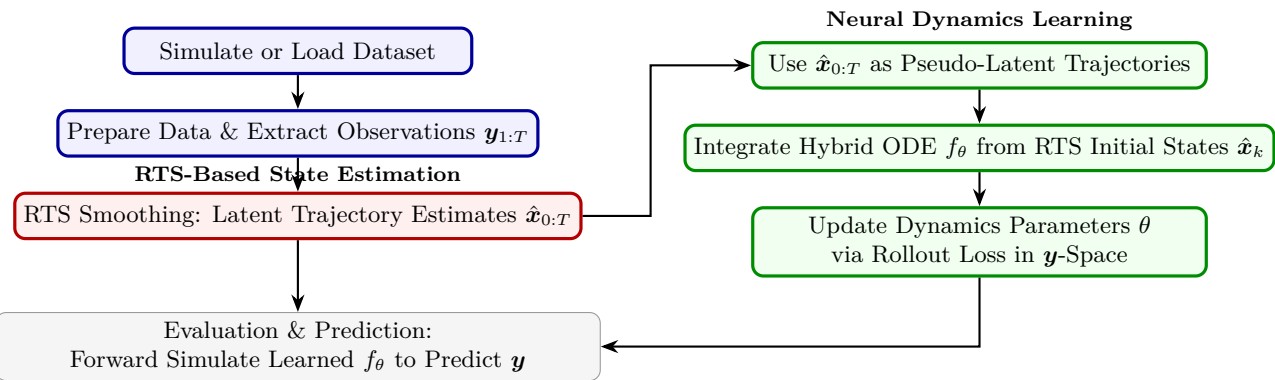

Figure 1: Overview of the proposed pipeline.

model expands as

$$p\big(\boldsymbol{y}_{1:T_i}^{(i)} \mid \boldsymbol{x}_{0:T_i}^{(i)}\big) = \prod_{k=1}^{T_i} p\big(\boldsymbol{y}_k^{(i)} \mid \boldsymbol{x}_k^{(i)}\big). \tag{4}$$

For the state model, $\boldsymbol{x}_k$ depends only on $\boldsymbol{x}_{k-1}$ through $\boldsymbol{f}_\theta$ (the Markov property), so the state-sequence density expands as

$$p\big(\boldsymbol{x}_{0:T_i}^{(i)}\big) = p\big(\boldsymbol{x}_0^{(i)}\big) \prod_{k=1}^{T_i} p\big(\boldsymbol{x}_k^{(i)} \mid \boldsymbol{x}_{k-1}^{(i)}\big). \tag{5}$$

Substituting eqs. (4) and (5) into the decomposition above gives the full Markov factorization:

$$p\big(\boldsymbol{x}_{0:T_i}^{(i)}, \boldsymbol{y}_{1:T_i}^{(i)}\big) = \underbrace{p\big(\boldsymbol{x}_0^{(i)}\big)}_{\text{prior}} \underbrace{\prod_{k=1}^{T_i} p\big(\boldsymbol{x}_k^{(i)} \mid \boldsymbol{x}_{k-1}^{(i)}\big)}_{\text{state transitions}} \underbrace{\prod_{k=1}^{T_i} p\big(\boldsymbol{y}_k^{(i)} \mid \boldsymbol{x}_k^{(i)}\big)}_{\text{measurements}}. \tag{6}$$

Since all noise is Gaussian, each factor contributes a precision-weighted quadratic to the negative log-likelihood: the measurement terms are weighted by $R^{-1}$ and the state-transition terms are weighted by $Q^{-1}$. The full closed-form expression is derived in section 3.

The difficulty with directly optimizing eq. (2) is that the state-model terms require access to the latent states $\boldsymbol{x}_{0:T_i}^{(i)}$, which are not measured. We do not have the true state sequence, nor do we have direct noisy measurements of every state variable. However, we can obtain an estimate of the full state sequence from a smoother that combines the available measurements with the current dynamics model. We therefore replace the unknown latent trajectory with the smoothed estimate $\hat{\boldsymbol{x}}_{0:T_i}^{(i)}$ produced by an RTS smoother, and treat the state-transition terms as if the smoothed states were the true states. This is a modeling assumption: each transition density $p(\boldsymbol{x}_k \mid \boldsymbol{x}_{k-1})$ is evaluated at the smoother estimates rather than the unknown ground truth, and the associated uncertainty is captured by the smoother covariance $P_k$ rather than the single-step process noise $Q$. The resulting surrogate objective, together with the measurement-model terms evaluated at the predicted observations, forms the basis of our training loss. The full formulation is given in section 3.3, and the smoother procedure is described in section 3.1 and appendix A.

## 2 Related Work

Integrating machine learning (ML) with Ordinary Differential Equations (ODEs) has been extensively explored to overcome the limitations of traditional dynamical modeling, particularly in scenarios with incomplete or noisy data. Neural ODEs Chen et al. (2018) have emerged as a popular method that enables continuous-time modeling directly from measuremental data. Extensions of this framework, such as Latent

ODEs Rubanova et al. (2019) and GRU-ODE-Bayes De Brouwer et al. (2019), incorporate recurrent neural network structures to handle irregularly sampled and partially measured time series, achieving notable success in physiological and clinical applications.

However, many of these approaches rely on neural encoders or recurrent architectures to infer latent states, which may struggle with generalization, interpretability, and stability, especially under sparse or highly noisy conditions Rubanova et al. (2019); De Brouwer et al. (2019). For instance, the Latent ODE framework Rubanova et al. (2019) learns a variational encoder to infer initial latent states but does not explicitly incorporate known structure from the underlying system into those latent states, potentially leading to physically implausible trajectories when data is limited or noisy. Similarly, GRU-ODE-Bayes De Brouwer et al. (2019) provides strong predictive capabilities but lacks explicit interpretability in its learned state transitions.

Hybrid modeling frameworks have sought to address these limitations by combining physically interpretable ODE structures with flexible neural components Rackauckas et al. (2020). For example, Universal Differential Equations Rackauckas et al. (2020) integrate neural networks directly into known mechanistic models, enabling the learning of unknown system terms while preserving interpretability. Bayesian filtering strategies for learning hybrid dynamics were introduced in Imbiriba et al. (2022; 2024), and subsequently extended to handle higher-order Markov dependencies Tang et al. (2024) and to enhance interpretability Straka et al. (2025). However, these approaches are largely restricted to a forward-filtering paradigm within an augmented state-space formulation, leading to computationally expensive and challenging optimization. Recent methods such as Neural Extended Kalman Filters Liu et al. (2024) and KalmanNet Revach et al. (2022) have introduced data-driven enhancements to classical Kalman filtering. KalmanNet employs recurrent neural networks to dynamically adjust Kalman gains, improving robustness in tracking tasks with partially known dynamics, yet it remains primarily focused on filtering rather than discovering unknown components of the underlying ODE. An in-depth discussion of AI-augmented Kalman-type algorithms can be found in Shlezinger et al. (2025).

More broadly, model-based deep learning methods such as KalmanNet and its unsupervised variants Revach et al. (2021) and the RTSNet smoother Revach et al. (2023) embed neural networks inside the flow of Kalman filtering or Rauch–Tung–Striebel smoothing. These approaches unfold classical estimators into trainable architectures that learn to filter or smooth in partially known, nonlinear state-space models, achieving strong performance under model mismatch and non-Gaussian noise. However, their end goal is still a learned *estimator* (filter/smoother) that must be run online at inference time; they are not primarily aimed at learning a hybrid neural differential equation that preserves the known structure of the dynamics while discovering the unknown parts. This is the main distinction from our approach, where the central object is the hybrid ODE itself, not the learned estimator.

(Demirkaya et al., 2021) introduced a preliminary hybrid ODE–NN framework that leverages the Cubature Kalman Filter (CKF) for joint hidden-state estimation and neural-parameter learning via recursive Bayesian estimation. This CKF-based training paradigm was later used and extended in (Demirkaya et al., 2024). While effective in partially observed physiological settings, CKF-based approaches can become computationally expensive as network depth and parameter dimension grow (due to covariance propagation and sigma-point evaluations), and typically rely on online filtering for state estimation during prediction, motivating further refinement toward more scalable and sequence-level inference.

Recognition ODEs Buisson-Fenet et al. (2023) address partial measurements by learning a recognition model, inspired by nonlinear observer theory, that maps measured outputs to the latent state. In that framework, latent-state estimation is handled by a learned observation-to-state map rather than by smoothing the full measurement sequence. In contrast, our method uses RTS smoothing to infer sequence-level latent trajectories and initial conditions, and uses these smoothed states directly to guide training. Similarly, recent hybrid neural-ODE approaches that incorporate symbolic regression Grigorian et al. (2024) or causal constraints Zou et al. (2024) emphasize interpretability and causality but do not explicitly address state estimation stability in noisy or sparse data settings.

Other methods, such as Whipple & Hernandez-Vargas (2024), explicitly augment the differential equation model with additional intermediate states so that partially unknown biological mechanisms can be repre-

sented within the dynamics itself. In that approach, these added states are treated as part of the system state, their trajectories are learned jointly with the observed components, and neural terms are used to model the corresponding unknown mechanisms. By contrast, our method uses RTS smoothing to estimate latent trajectories for the missing or unmeasured states in the chosen state-space model during training, and then learns the corresponding unknown dynamics within that model.

In contrast, the proposed method is designed to address three recurring limitations in prior work: (i) latent-state inference is often handled by learned encoders or recurrent modules that can be brittle under noisy or partial measurements; (ii) the resulting latent representations are often weakly tied to known system structure; and (iii) many methods either require online estimation at test time or do not produce a reusable dynamical model. Our approach addresses these issues by using RTS smoothing to estimate latent trajectories for training and then distilling this information into a standalone hybrid ODE model. The advantages of the proposed method over other methods include:

- **Robustness to Noise and Sparsity:** Kalman-style estimators propagate uncertainty through nonlinear system dynamics, offering smoothed trajectories that can be less sensitive to noisy measurements than purely neural inference modules.

- **Consistency with known dynamics:** By leveraging domain-specific structure in the state-space model, the estimator encourages latent trajectories that remain physically meaningful, guiding neural ODE training toward realistic dynamics.

- **Separation of state estimation and dynamics learning:** State estimation is decoupled from parameter learning, providing stable pseudo-target trajectories for training while the final deployed model is a compact physics-based neural differential equation that does not require online estimation for forward prediction.

The proposed framework thus combines the benefits of classical Kalman filtering (robustness, interpretability, stability) with the flexibility and power of neural ODE learning, while yielding a reusable dynamical model that can be simulated and generalized across subjects and systems. This significantly improves over existing hybrid approaches and estimator-focused methods in scenarios involving partial observability, measurement noise, and cross-subject variability.

## 3 METHODS

We consider stochastic dynamical systems with latent state $\boldsymbol{x}(t)$ and measurements $\boldsymbol{y}(t)$. We discretize the continuous-time model in eq. (1) as

$$\boldsymbol{x}_k = \boldsymbol{f}_\theta(\boldsymbol{x}_{k-1}, t_k) + \boldsymbol{w}_k \,, \tag{7}$$

where we use the shorthand $\boldsymbol{x}_k = \boldsymbol{x}(t_k)$ to make notation more compact. The *process noise* $\boldsymbol{w}_k$ is normally distributed with zero mean and covariance $Q$. We treat measurements as an instantaneous lower dimensional mapping of the state vector $\boldsymbol{x}(t_k)$ at time $t_k$

$$\boldsymbol{y}_k = \boldsymbol{h}(\boldsymbol{x}_k) + \boldsymbol{\varepsilon}_k \,, \tag{8}$$

where the *measurement noise* $\boldsymbol{\varepsilon}_k$ is normally distributed with zero mean and covariance $R$. The measurement error at $t_k$ is uncorrelated to the one at $t_l$ when $t_k \neq t_l$. The initial state $\boldsymbol{x}_0$, is normally distributed with mean $\bar{\boldsymbol{x}}_0$ and covariance $P_0$. The initial state, process noise, and measurement error are uncorrelated. We assume that the covariance matrices $Q$ and $R$ are time-independent. Being able to infer the state vector trajectory, i.e. $\boldsymbol{x}(t)$, from measurements not only enables researchers to forecast the system's behavior in the future, but also they can quantify the system's response if they were to intervene.

Unfortunately, in a realistic setting we have partial knowledge of eq. (7) leading to the following additional two challenges beyond that of estimating the trajectory of the state vector. The first challenge is that we often do not know the parametric form for a subset of the differential equations. The second challenge focuses on the case where we know the equations parametric form but do not know the values of their parameters.

To address the first challenge we approximate those unknown equations with neural networks, leaving us with the need to solve the second challenge that of parameter estimation.

We now write the exact joint log-likelihood in closed form using eq. (6). From eq. (7), $\boldsymbol{x}_k = \boldsymbol{f}_\theta(\boldsymbol{x}_{k-1}, t_k) + \boldsymbol{w}_k$ with $\boldsymbol{w}_k \sim \mathcal{N}(\boldsymbol{0}, Q)$, so the transition density is $\boldsymbol{x}_k \mid \boldsymbol{x}_{k-1} \sim \mathcal{N}(\boldsymbol{f}_\theta(\boldsymbol{x}_{k-1}, t_k), Q)$, contributing a $Q^{-1}$-weighted quadratic to the log-likelihood. From eq. (8), $\boldsymbol{y}_k = \boldsymbol{h}(\boldsymbol{x}_k) + \boldsymbol{\varepsilon}_k$ with $\boldsymbol{\varepsilon}_k \sim \mathcal{N}(\boldsymbol{0}, R)$, so the measurement density is $\boldsymbol{y}_k \mid \boldsymbol{x}_k \sim \mathcal{N}(\boldsymbol{h}(\boldsymbol{x}_k), R)$, contributing an $R^{-1}$-weighted quadratic. Substituting into eq. (6) and dropping constants that do not depend on $\theta$ or $\boldsymbol{x}_{0:T}$, we obtain

$$
\log \Pr(\boldsymbol{x}_{0:T}, \boldsymbol{y}_{1:T}) = -\frac{1}{2} \Bigg( \log \det(P_0) + (\boldsymbol{x}_0 - \bar{\boldsymbol{x}}_0)^\mathsf{T} P_0^{-1} (\boldsymbol{x}_0 - \bar{\boldsymbol{x}}_0)
$$
$$
+ T \cdot \log \det(Q) + \sum_{k=1}^T (\boldsymbol{x}_k - \boldsymbol{f}_\theta(\boldsymbol{x}_{k-1}, t_k))^\mathsf{T} Q^{-1} (\boldsymbol{x}_k - \boldsymbol{f}_\theta(\boldsymbol{x}_{k-1}, t_k))
$$
$$
+ T \cdot \log \det(R) + \sum_{k=1}^T (\boldsymbol{y}_k - \boldsymbol{h}(\boldsymbol{x}_k))^\mathsf{T} R^{-1} (\boldsymbol{y}_k - \boldsymbol{h}(\boldsymbol{x}_k)) \Bigg). \tag{9}
$$

The three lines correspond directly to the three factors in eq. (6): the prior on $\boldsymbol{x}_0$ (weighted by $P_0^{-1}$), the process model (weighted by $Q^{-1}$), and the measurement model (weighted by $R^{-1}$). In each case, the inverse covariance acts as a natural precision weight: terms with larger uncertainty contribute less to the log-likelihood.

As discussed in section 1.1, we cannot directly optimize eq. (9) because the state-transition terms require the latent states $\boldsymbol{x}_{0:T}$, which are not observed. We therefore construct a surrogate objective by replacing the unknown latent trajectory with the smoothed estimate $\hat{\boldsymbol{x}}_{0:T}$ produced by an RTS smoother (section 3.1). Under this substitution, each state-transition term $p(\boldsymbol{x}_k \mid \boldsymbol{x}_{k-1})$ is evaluated at the smoother estimates rather than the unknown ground truth. The smoother also provides a covariance $P_k$ at each time step that quantifies its confidence in $\hat{\boldsymbol{x}}_k$. Since $P_k$ captures the accumulated information from the entire measurement sequence rather than a single transition, we use the smoother covariance in place of the single-step process-noise covariance $Q$ when weighting the state-transition errors. This reweighting is a modeling choice rather than an algebraic consequence of the substitution: the resulting objective is no longer the exact MLE, but a smoother-guided surrogate in which residuals are weighted by posterior confidence rather than the single-step transition uncertainty $Q$. The objective optimized in practice is therefore the surrogate training loss introduced below, rather than the exact maximum-likelihood objective in eq. (9).

Concretely, fixing the latent trajectory to the smoothed values and examining what remains in eq. (9), the prior term (first line) does not depend on $\theta$. The process-model term (second line) becomes a weighted squared error between the smoothed state and the one-step model prediction. The measurement-model term (third line) penalizes the mismatch between predicted and observed measurements, weighted by $R^{-1}$. For the horizon-1 case, this yields the surrogate loss

$$
\mathcal{L} = \frac{1}{T} \sum_{k=1}^T \left[ (\hat{\boldsymbol{x}}_k - \tilde{\boldsymbol{x}}_k)^\mathsf{T} P_k^{-1} (\hat{\boldsymbol{x}}_k - \tilde{\boldsymbol{x}}_k) \; + \; (\boldsymbol{y}_k - \boldsymbol{h}(\tilde{\boldsymbol{x}}_k))^\mathsf{T} R^{-1} (\boldsymbol{y}_k - \boldsymbol{h}(\tilde{\boldsymbol{x}}_k)) \right], \tag{10}
$$

where $\tilde{\boldsymbol{x}}_k$ is the state at $t_k$ obtained by integrating the dynamical system forward one step from the initial condition $\hat{\boldsymbol{x}}_{k-1}$. We define prediction horizons to be time gap between the initial condition and the predicted state after integration. In eq. (10), the prediction horizon $n = 1$. In section 3.2, we discuss why restricting the objective to the $n = 1$ horizon can be insufficient motivating the use of $n > 1$ horizons. The first term is the state-space error weighted by the smoother precision $P_k^{-1}$, and the second term is the measurement-space error weighted by $R^{-1}$. Using $P_k^{-1}$ instead of $Q^{-1}$ accounts for the uncertainty in each smoothed state estimate: if the smoother is confident about $\hat{\boldsymbol{x}}_k$ (small $P_k$), the corresponding error is weighted more heavily; if it is less confident (large $P_k$), the error is downweighted.

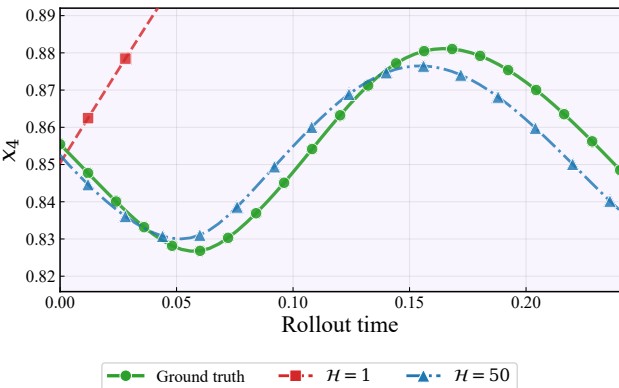

Figure 2: Effect of integration horizon on yeast glycolysis rollout for the latent state $x_4$. A single-step objective ($\mathcal{H} = 1$) is not sufficient to maintain accurate long-horizon rollouts, leading to early drift from the ground truth. In contrast, training with a longer horizon ($\mathcal{H} = 50$) produces a trajectory that remains closer to the reference.

### 3.1 RTS-guided state estimation

Directly optimizing over unconstrained latent trajectories is ill-posed under partial observability: many latent paths can explain the same measurements while violating the system dynamics. To restrict the search to physically and temporally consistent trajectories, we replace the free latent variables with structured estimates produced by a RTS smoother:

$$\hat{\boldsymbol{x}}_{0:T_i}^{(i)}(\theta),\ \mathbf{P}_{0:T_i}^{(i)}(\theta)\ =\ \mathcal{S}_\phi\big(\boldsymbol{y}_{1:T_i}^{(i)};\,\boldsymbol{f}_\theta,\,\boldsymbol{h}\big), \tag{11}$$

where $\mathcal{S}_\phi(\cdot)$ denotes the proposed procedure with fixed hyperparameters $\phi$ (noise covariances $Q$, $R$, and initialization settings). Given $\boldsymbol{f}_\theta$, the smoother deterministically maps the full measurement sequence to a latent trajectory that is consistent with both the assumed noise model and the current dynamics. In particular, the smoothed initial state $\hat{\boldsymbol{x}}_0^{(i)}(\theta)$ is not a free variable but is inferred from the full sequence as the first component of the smoothed trajectory. The detailed CKF and RTS recursions are given in appendix A.

### 3.2 Impact of integration horizon

The loss in eq. (10) compares the integrated state to the smoother estimate over one step. This gives a useful local training signal, but it does not by itself control what happens over a longer rollout. Prior work has noted that step by step prediction error can accumulate over long rollouts (Venkatraman et al., 2015; Asadi et al., 2018). This is especially important in the partially observed setting considered here. When some state variables are not measured directly, an error in the learned dynamics of those variables may have only a small effect after one step. The same error can appear more clearly only after several integration steps, once it propagates into the measured variables. As a result, a model can look reasonable under a one-step loss and still drift over time. This effect is shown in fig. 2. The model trained with $\mathcal{H} = 1$ departs from the ground-truth trajectory early in the rollout, while the model trained with a longer horizon stays closer over the same window. This is also consistent with recent results showing that under misspecification due to partial observability, multi-step prediction can be preferable to single-step training (Somalwar et al., 2025). For these reasons, we include rollout errors at longer horizons in the training objective instead of training only with $n = 1$.

### 3.3 Multi-horizon training objective

Using the smoothed states, we generate multi-step predictions by integrating the dynamics forward:

$$\tilde{\boldsymbol{x}}_{k+n}^{(i,n)}(\theta) = \hat{\boldsymbol{x}}_k^{(i)}(\theta) + \int_{t_k}^{t_{k+n}} \boldsymbol{f}_\theta(\boldsymbol{x}(t))\, dt, \qquad \tilde{\boldsymbol{y}}_{k+n}^{(i,n)}(\theta) = \boldsymbol{h}\big(\tilde{\boldsymbol{x}}_{k+n}^{(i,n)}(\theta)\big), \tag{12}$$

for prediction horizons $n$ in a finite set $\mathcal{H}$ (e.g., single-step and one or more longer time scales). For each horizon we measure both a latent-state error and an observation-space error:

$$\boldsymbol{e}_{k+n}^{(x,i,n)}(\theta) = \tilde{\boldsymbol{x}}_{k+n}^{(i,n)}(\theta) - \hat{\boldsymbol{x}}_{k+n}^{(i)}(\theta), \qquad \boldsymbol{e}_{k+n}^{(y,i,n)}(\theta) = \tilde{\boldsymbol{y}}_{k+n}^{(i,n)}(\theta) - \boldsymbol{y}_{k+n}^{(i)}. \tag{13}$$

The full training objective generalizes eq. (10) to multiple horizons and sequences:

$$\theta^\star = \arg\min_\theta \ \frac{1}{T_{\text{tot}}} \sum_{i=1}^N \sum_{k=1}^{T_i} \sum_{n \in \mathcal{H}} \left( \big\| \boldsymbol{e}_{k+n}^{(y,i,n)}(\theta) \big\|_{\mathbf{W}^{(y)}}^2 \ + \ \big\| \boldsymbol{e}_{k+n}^{(x,i,n)}(\theta) \big\|_{\mathbf{W}_{k+n}^{(x,i)}(\theta)}^2 \right), \tag{14}$$

where $\|\boldsymbol{e}\|_{\mathbf{W}}^2 := \boldsymbol{e}^\top \mathbf{W} \boldsymbol{e}$. The observation-space weight mirrors the $R^{-1}$ term in eq. (10):

$$\mathbf{W}^{(y)} = R^{-1}, \tag{15}$$

so that measurements with larger noise variance contribute less to the loss. The latent-state weight uses the smoother covariance, as in eq. (10):

$$\mathbf{W}_{k+n}^{(x,i)}(\theta) = \big(\mathbf{P}_{k+n}^{(i)}(\theta)\big)^{-1}, \tag{16}$$

so that time steps where the smoother is uncertain (large $\mathbf{P}_{k+n}^{(i)}$) have a smaller influence on the latent-state term. The summation over horizons $\mathcal{H}$ enforces consistency of the learned dynamics across multiple prediction time scales. When $\mathcal{H} = \{1\}$, eq. (14) reduces to the single-step surrogate loss in eq. (10) summed over sequences.

### 3.4 Alternating optimization

The RTS smoother plays a dual role: it provides initial states for integration and produces latent trajectories that regularize learning. Both the parameters $\theta$ and the smoothed trajectories $\hat{\boldsymbol{x}}_{0:T_i}^{(i)}$ depend on each other through the smoother. We exploit this structure with an alternating optimization scheme. Given $\theta^{(m)}$:

1. **State-estimation step:** Run the RTS smoother $\mathcal{S}_\phi$ with dynamics $\boldsymbol{f}_{\theta^{(m)}}$ to obtain updated trajectories $\hat{\boldsymbol{x}}_{0:T_i}^{(i)}(\theta^{(m)})$ and covariances $\mathbf{P}_k^{(i)}(\theta^{(m)})$ for all sequences and time steps.

2. **Parameter-update step:** Treat the smoothed trajectories and covariances as fixed and update $\theta$ to $\theta^{(m+1)}$ by (stochastic) gradient descent on eq. (14).

Iterating these steps alternately refines the latent trajectories, their uncertainties, and the dynamics parameters.

### 3.5 Hybrid neural-physics transition model

In all experiments we model the latent dynamics with a hybrid ODE that combines known mechanistic equations with neural networks that replace unknown parts of the system. We partition the state as $\boldsymbol{x}(t) = \big(\boldsymbol{x}_{\text{phys}}(t), \boldsymbol{x}_{\text{unk}}(t)\big)$, where $\boldsymbol{x}_{\text{phys}}$ collects states whose dynamics are specified analytically, while $\boldsymbol{x}_{\text{unk}}$ denotes the subset of states whose dynamics are not reliably specified and are therefore parameterized by neural networks.

For state variables with known dynamics, we keep the original ODE terms,

$$\dot{\boldsymbol{x}}_{\text{phys}}(t) = f_{\text{phys}}\bigg(\boldsymbol{x}_{\text{phys}}(t),\, \boldsymbol{x}_{\text{unk}}(t)\bigg),$$

while for states with missing dynamics we introduce a neural network that provides the entire right-hand side,

$$\dot{\boldsymbol{x}}_{\text{unk}}(t) = f_{\text{unk}}\bigg(\boldsymbol{x}_{\text{phys}}(t),\ \boldsymbol{x}_{\text{unk}}(t),\ \boldsymbol{\theta}\bigg),$$

where $f_{\text{unk}}$ is parameterized by a small MLP (typically $2-4$ layers with Tanh or ReLU activations). The full hybrid dynamical system is then

$$\dot{\boldsymbol{x}}(t) = \boldsymbol{f}\big(\boldsymbol{x}(t),\boldsymbol{\theta}\big) := \bigg(f_{\text{phys}}\big(\boldsymbol{x}_{\text{phys}}(t),\ \boldsymbol{x}_{\text{unk}}(t)\big),\ f_{\text{unk}}\big(\boldsymbol{x}_{\text{phys}}(t),\ \boldsymbol{x}_{\text{unk}}(t),\ \boldsymbol{\theta}\big)\bigg). \tag{17}$$

This construction makes explicit that known components remain as analytical ODEs (which may depend on both $\boldsymbol{x}_{\text{phys}}$ and $\boldsymbol{x}_{\text{unk}}$), while states with unknown dynamics are governed entirely by learned neural ODE terms.

## 4 Experiments and Results

### 4.1 Choice of Benchmark Dynamical Systems

We evaluate the proposed methodology on five different dynamical systems that are familiar in the ML literature on neural differential equations, but varied enough to exercise the main difficulties in this paper: learning missing dynamics from partial measurements and maintaining stable long-horizon behavior. The suite includes small diagnostic examples alongside higher-dimensional and stiff models where training can become fragile.

**Harmonic oscillator.** A minimal linear baseline with one unmeasured state. It is mainly used to check latent-state reconstruction and to verify that the learned hybrid dynamics reproduce the expected phase-plane structure.

**Hodgkin–Huxley neuron.** A stiff, multi-state biophysical model with several unmeasured gating variables. It is a useful stress test because small errors in these latent states can strongly affect the measured voltage trajectory.

**Retinal circulation.** A physiological model with indirect measurements and nonlinear pressure–flow interactions. This benchmark is closer to the partially measured settings that motivate hybrid modeling in practice.

**Brusselator reaction model.** A compact nonlinear reaction network with oscillatory behavior and strong coupling. It provides a clean setting for checking whether the learned dynamics capture limit-cycle structure when only a subset of species is measured.

**Yeast glycolysis oscillator.** A higher-dimensional biochemical oscillator with dense coupling and one explicitly unmeasured state. Compared with the Brusselator, it probes whether the approach remains stable as the latent network becomes more complex while still being fully specified and reproducible.

Across all benchmarks we follow the same modeling template: we keep the known mechanistic terms explicit and add neural components to states where the dynamics are missing. Model and architecture details for each dataset are provided in Appendix B.

### 4.2 Setup and baselines

We evaluate the proposed method on the partially observed dynamical systems mentioned in section 4.1. In each system, only a subset of state variables is measured; the remaining variables are latent and must be inferred. We compare the proposed method against four existing approaches for learning dynamics from partial and noisy measurements. NeuralODE (Chen et al., 2018) is the standard neural differential equation model and tests how far a plain continuous-time vector field can go without an explicit state estimator. GRU-ODE-Bayes (De Brouwer et al., 2019) augments Neural ODE dynamics with GRU-style updates at observation times together with a Bayesian update rule, providing a strong learned-inference baseline

for irregular and partially observed sequences. We also compare against Recognition ODE / structured NODE (Buisson-Fenet et al., 2023), which addresses partial observability by coupling a structured Neural ODE with a learned recognition model that maps observation histories to latent states. Finally, we include a Cubature Kalman Filter (CKF) based hybrid ODE–NN approach (Demirkaya et al., 2021) that combines mechanistic ODE structure with neural components, but performs state and parameter estimation through recursive Bayesian estimation (RBSE; online filtering) rather than backpropagation-based training.

These baselines provide standard reference points for the main goal of this paper: learning dynamics when some state variables are never measured. A plain NeuralODE does not by itself infer the missing state or full initial condition from measurements and therefore requires an additional inference mechanism. GRU-ODE-Bayes and Recognition ODE address this by learning inference modules that incorporate observations over time, while the CKF-based hybrid ODE–NN baseline estimates hidden states through forward recursive Bayesian filtering. In contrast, the proposed approach uses smoothing-based state estimation to produce dynamics-consistent *smoothed* latent trajectories and covariances as training guidance, then distills this information into a standalone hybrid ODE that can be forward simulated at test time without filtering, smoothing, or a learned recognition model.

We evaluate the proposed method using two complementary error metrics that target different aspects of the learned dynamics. The first metric we use captures the state estimation error and is calculated using the root-mean-square error (RMSE) between the *unmeasured* latent components $\boldsymbol{x}_{\text{unk}}$ and the true state

$$\text{RMSE}_{\text{unk}}^{(m)} = \sqrt{\frac{1}{T_{\text{tot}}d_{\text{unk}}} \sum_{i=1}^{N} \sum_{k=1}^{T_i} \left\| \hat{\boldsymbol{x}}_{\text{unk},k}^{(i,m)} - \boldsymbol{x}_{\text{unk},k}^{(i,m)} \right\|_2^2}, \tag{18}$$

where $m = 1, \ldots, M$ is the index of the Monte Carlo simulation. We compute and report the average over $M = 50$ simulations,

$$\overline{\text{RMSE}}_{\text{unk}} = \frac{1}{M} \sum_{m=1}^{M} \text{RMSE}_{\text{unk}}^{(m)}. \tag{19}$$

Note that during training and simulation the model does *not* use measurements of these components; the ground-truth $\boldsymbol{x}_{\text{unk},k}^{(i,m)}$ is used only for offline evaluation, to quantify how well the method can reconstruct a missing state for which no direct measurements would be available in practice.

The second metric we use is the Hausdorff distance between the true and the integrated state trajectories. The Hausdorff distance captures the state-space trajectory error. For each dynamical system we form the corresponding point sets

$$\mathcal{X} = \{x_k | k = 0, \ldots, T\}, \ \tilde{\mathcal{X}} = \{\tilde{x}_k | k = 0, \ldots, T\}. \tag{20}$$

where $\mathcal{X}$ is the true state trajectory. $\hat{\mathcal{X}}$ is the state trajectory computed by integrating the hybrid dynamical system using the estimated parameters $\boldsymbol{\theta}$. We use the same initial condition in both cases. The Hausdorff distance between these sets is

$$d_H(\mathcal{X}, \hat{\mathcal{X}}) := \max\left( \sup_{\boldsymbol{x} \in \mathcal{X}} d(\boldsymbol{x}, \tilde{\mathcal{X}}), \sup_{\tilde{\boldsymbol{x}} \in \tilde{\mathcal{X}}} d(\mathcal{X}, \tilde{\boldsymbol{x}}) \right), \tag{21}$$

where

$$d(\boldsymbol{a}, B) = \inf_{\boldsymbol{b} \in B} \left( |\boldsymbol{a} - \boldsymbol{b}|^2 \right). \tag{22}$$

Unlike pointwise time-aligned errors, the Hausdorff distance compares trajectories as geometric objects and as a consequence it better captures the similarity between two state trajectories. To emphasize this point imagine two identical timeseries with the second one being shifted in time by a small amount. Even though the two trajectories have identical features they will end up having a high RMSE since it is a pointwise comparison metric as we can see in eq. (18).

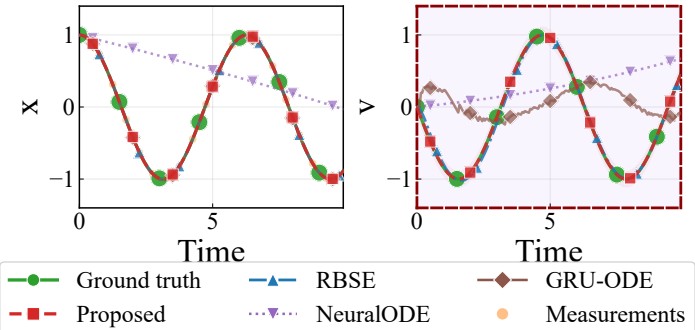

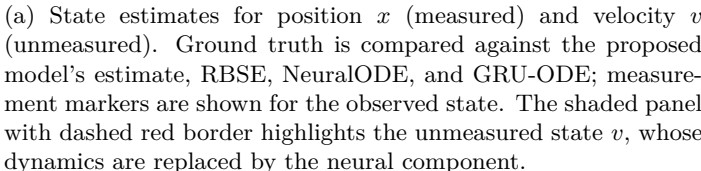

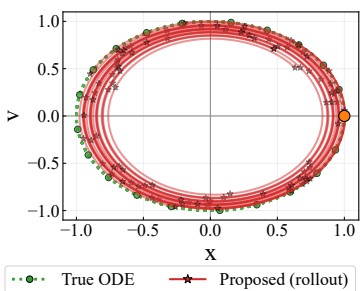

(a) State estimates for position $x$ (measured) and velocity $v$ (unmeasured). Ground truth is compared against the proposed model's estimate, RBSE, NeuralODE, and GRU-ODE; measurement markers are shown for the observed state. The shaded panel with dashed red border highlights the unmeasured state $v$, whose dynamics are replaced by the neural component.

(b) Estimated phase portrait ($x$ vs. $v$). The dotted curve shows the ground-truth trajectory, while the solid curve shows the phase portrait formed by the proposed state estimates. Only position $x$ is measured, and velocity $v$ is reconstructed through RTS guidance.

Figure 3: Harmonic oscillator state estimation results. (a) Time-series estimates for the measured state $x$ and the unmeasured state $v$. (b) State-space trajectory showing recovery of the oscillator geometry.

## 4.3 Qualitative results

### 4.3.1 Harmonic Oscillator

In the harmonic oscillator experiment, only the position $x$ is observed, while the velocity $v$ remains unmeasured. This provides a simple test case for the missing-state setting considered in this work: the model must recover the latent velocity from noisy position measurements and use that information to learn the unknown part of the dynamics.

Figure 3 shows the resulting state estimates and phase portrait. The proposed method recovers the latent velocity trajectory accurately from the partial observations and yields state estimates that remain consistent with the underlying oscillator dynamics. In phase space, the learned trajectory preserves the expected closed-orbit structure, indicating that the model captures not only the observed position signal but also the coupled latent dynamics.

Overall, this experiment illustrates the main advantage of the proposed approach in a controlled setting: when part of the state is never observed directly, RTS-guided training can still recover the missing state and produce a learned dynamical model with the correct qualitative behavior. A representative rollout for the latent velocity $v$ is also included in the cross-system comparison in Figure 9.

### 4.3.2 Yeast Glycolysis

In the yeast glycolysis oscillator, all states except $x_4$ are measured. The proposed framework uses RTS smoothing to reconstruct the latent trajectory of $x_4$ and uses these smoothed states to guide training of the hybrid ODE. The proposed method captures the oscillatory behavior of the system and improves estimation of the unobserved state $x_4$ relative to the CKF-only baseline. The recovered trajectories also preserve the phase relationships among the measured metabolites. Figure 4 shows the RTS-smoothed state estimates together with ground truth. The proposed method closely follows the underlying oscillatory structure and produces smoother, more consistent estimates than the CKF-only baseline. A representative rollout for the unmeasured metabolite $x_4$ is shown together with the other benchmark systems in Figure 9.

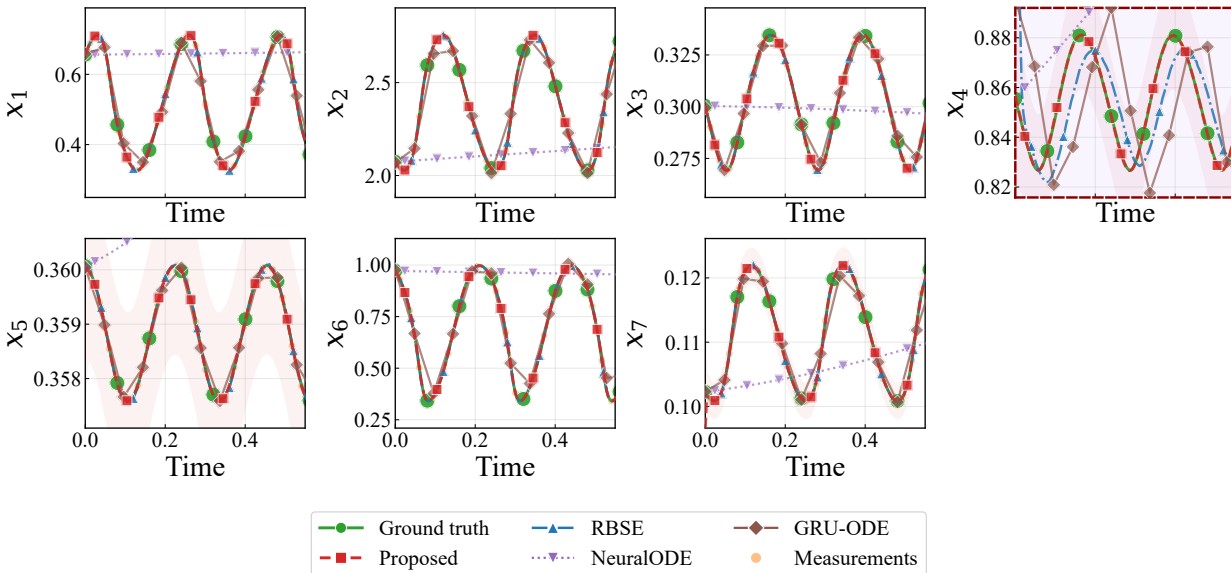

Figure 4: Yeast glycolysis state estimates. RTS-smoothed estimates from the proposed method (dashed, with $\pm 2\sigma$ uncertainty band) are shown against ground truth (solid); measurements are shown for observed states. The shaded panel with dashed red border highlights the intermediate state $x_4$, which is unmeasured and whose dynamics are replaced by the neural component.

### 4.3.3 Retinal Circulation

In the retinal circulation model, capillary pressure $P_4$ is not directly measured. The hybrid model replaces the corresponding compartment dynamics with a neural component while retaining the remaining mechanistic pressure equations. Using RTS-smoothed trajectories during training allows the model to infer the latent pressure dynamics and learn consistent pressure–flow coupling. The resulting hybrid model produces stable rollouts that reproduce the qualitative relationships among arterial, capillary, and venous pressures. Figure 5 shows rollout time-series predictions. The proposed method captures the overall coupling between pressure states and maintains realistic dynamical behavior over extended simulation horizons. A representative rollout for the latent capillary pressure $P_4$ is included in the combined comparison shown in Figure 9.

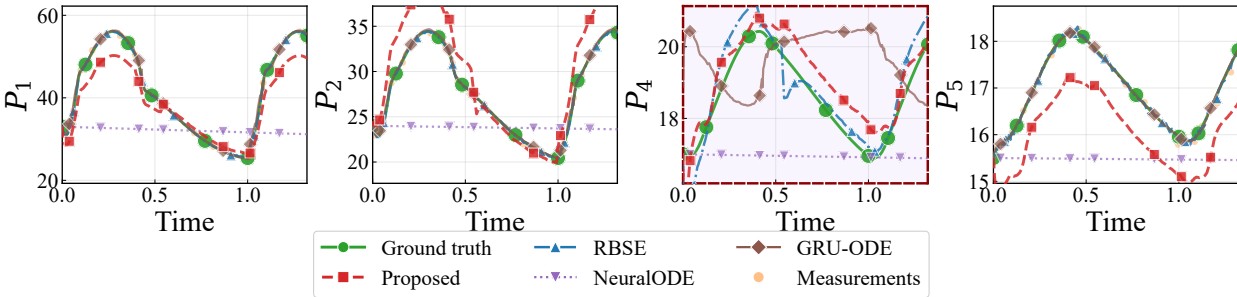

Figure 5: Retinal circulation state estimations for pressures $P_1$, $P_2$, $P_4$, and $P_5$. Ground truth is compared with state estimates from the proposed hybrid model, RBSE, NeuralODE, and GRU-ODE; measurements are shown for observed states. The shaded panel with dashed red border highlights the latent capillary pressure $P_4$, whose dynamics are replaced by the neural component and which is inferred during training.

### 4.3.4 Brusselator

The Brusselator reaction model provides a nonlinear oscillatory system in which one chemical species is unmeasured. RTS smoothing enables the model to reconstruct this latent species from the observed components and guides the training of the hybrid ODE.

The proposed method successfully recovers the limit-cycle structure of the system and produces rollouts that remain close to the ground-truth trajectory. Compared with NeuralODE and CKF-only baselines, the learned hybrid dynamics better preserve the oscillation geometry in phase space. A representative rollout for the unmeasured species $x$ is also shown in the cross-system summary in Figure 9. Figure 6 shows the estimated state trajectories.

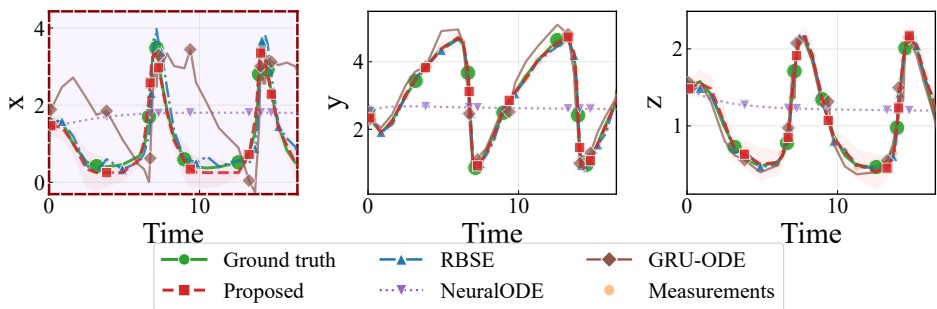

Figure 6: Brusselator state-estimation time series. Ground truth (solid) and noisy measurements (orange markers; observed components) are shown along with proposed RTS-smoothed estimates (dashed, with $\pm 2\sigma$ band). The shaded panel with dashed red border highlights the unmeasured species $x$, whose dynamics are replaced by the neural component.

### 4.3.5 Hodgkin–Huxley Neuron

For the Hodgkin–Huxley neuron model, the membrane voltage $V$ and gating variables $(h, n)$ are observed, while the $m$-gate remains latent. In the hybrid model we replace the dynamics of the $m$-gate with a neural component and retain the remaining mechanistic equations. The RTS smoother reconstructs the hidden $m$ trajectory from the observed components, providing dynamics-consistent pseudo-targets for training. The learned model reproduces realistic voltage spiking behavior and captures the coupling between $V$ and the gating variables through the recovered $m$-dynamics. Figure 7 shows the reconstructed state trajectories, while Figure 8 presents state-space trajectories involving $V$ and gating variables. The proposed model produces trajectories that align closely with the ground-truth dynamics and preserves the characteristic

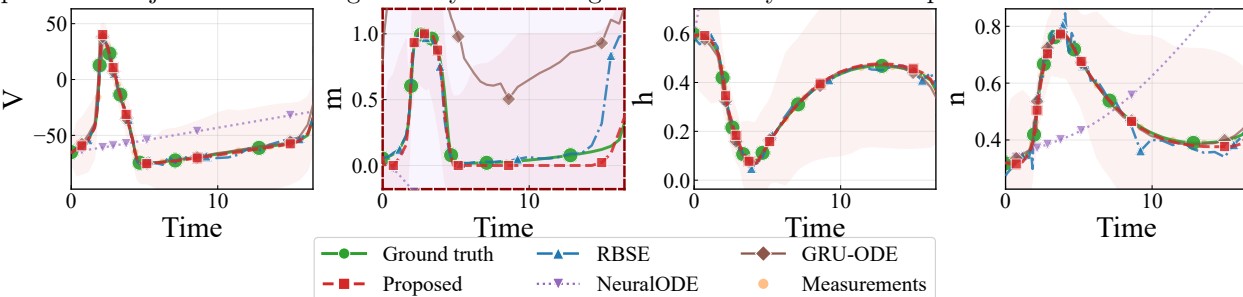

Figure 7: Hodgkin–Huxley time series under partial measurements. Proposed RTS-smoothed state estimates (dashed, with $\pm 2\sigma$ band) and CKF-only baseline estimates (dash-dot) are compared to ground truth (solid); measurements (orange markers) are available for $V, h, n$ while the $m$ gate is unobserved and must be inferred.

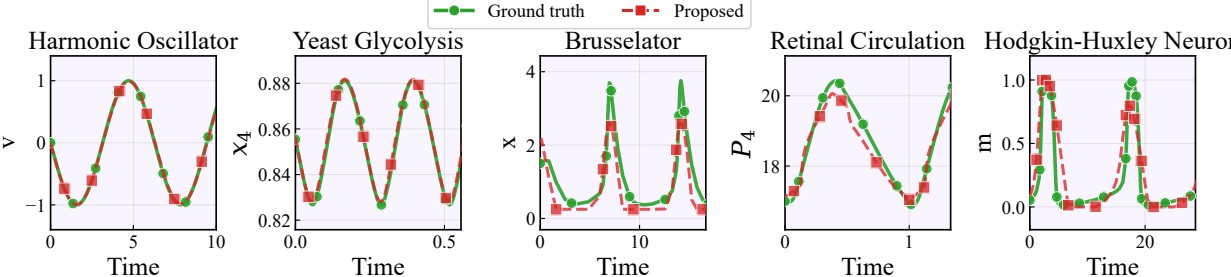

Figure 9: Representative rollout comparison for the unmeasured state in five benchmark systems: velocity $v$ in the harmonic oscillator, metabolite $x_4$ in yeast glycolysis, species $x$ in the Brusselator, capillary pressure $P_4$ in retinal circulation, and gating variable $m$ in Hodgkin–Huxley. Each panel shows the best selected prediction over the displayed time window. Ground truth is shown in green and the proposed trajectory in red.

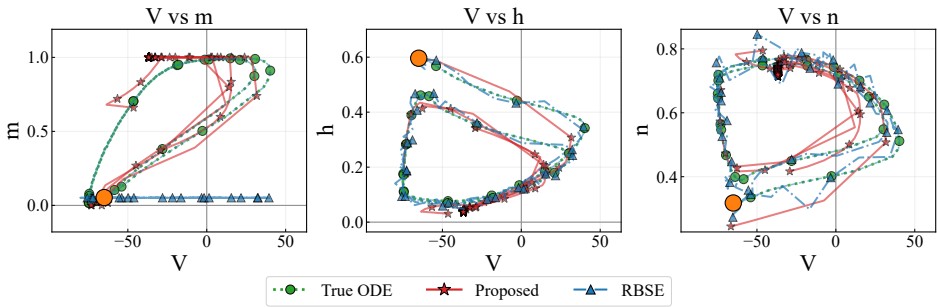

Figure 8: Hodgkin–Huxley state-space trajectories ($V$ versus gating variables). Dotted curve shows the true trajectory, while the proposed hybrid model trajectory (solid) is initialized from inferred states; agreement indicates recovery of the latent $m$-gate dynamics and spike-related state geometry.

### 4.4 Comparison of rollout behavior across benchmark systems

We next compare rollout behavior across the five benchmark systems using the unmeasured state in each model: velocity $v$ for the harmonic oscillator, metabolite $x_4$ for yeast glycolysis, species $x$ for the Brusselator, capillary pressure $P_4$ for retinal circulation, and the gating variable $m$ for Hodgkin–Huxley. Figure 9 summarizes representative predictions of the proposed method against ground truth over the displayed time window. Across systems, the learned dynamics remain close to the reference trajectories and preserve the main qualitative behavior of the latent state.

### 4.5 Quantitative results

Tables 1 and 2 report results on latent-state RMSE for the unmeasured states and rollout Hausdorff distance. On latent-state RMSE, the proposed method achieves the lowest error on four of the five benchmark systems, with especially large gains on yeast glycolysis and the harmonic oscillator. For example, on yeast glycolysis the RMSE drops to $4.08 \times 10^{-4}$, compared with $1.92 \times 10^{-1}$ for NeuralODE, $3.21 \times 10^{-2}$ for GRU-ODE-Bayes, and $1.71 \times 10^{-2}$ for the RBSE. The main exception is retinal circulation, where the RBSE attains a slightly lower latent-state RMSE.

The rollout results show a similarly favorable pattern. The proposed method achieves the lowest Hausdorff distance on all five systems, indicating better agreement with the geometry of the ground-truth trajectories over long horizons. The improvement is particularly clear on yeast glycolysis and the Brusselator, and remains competitive on retinal circulation even though that system favors the CKF/RBSE hybrid on latent-state RMSE. Taken together, these results indicate that RTS-guided training improves hidden-state reconstruction

and leads to more accurate learned dynamics under partial observability. The qualitative rollout comparison in Figure 9 complements the quantitative results.

Table 1: State-estimate RMSE (unmeasured states only).

| System | Proposed | NeuralODE (Chen et al., 2018) | GRU-ODE-Bayes (De Brouwer et al., 2019) | CKF/RBSE Hybrid (Demirkaya et al., 2021) |
|---|---|---|---|---|
| Harmonic Oscillator | $\mathbf{3.65 \times 10^{-4}}$ | $9.31 \times 10^{-1}$ | $7.55 \times 10^{-1}$ | $2.55 \times 10^{-2}$ |
| Hodgkin–Huxley Neuron | $\mathbf{5.18 \times 10^{-2}}$ | $4.97 \times 10^{0}$ | $7.89 \times 10^{-1}$ | $1.04 \times 10^{-1}$ |
| Brusselator | $\mathbf{2.33 \times 10^{-1}}$ | $1.25 \times 10^{0}$ | $1.57 \times 10^{0}$ | $3.53 \times 10^{-1}$ |
| Yeast Glycolysis | $\mathbf{4.08 \times 10^{-4}}$ | $1.92 \times 10^{-1}$ | $3.21 \times 10^{-2}$ | $1.71 \times 10^{-2}$ |
| Retinal Circulation | $8.31 \times 10^{-1}$ | $2.22 \times 10^{0}$ | $2.18 \times 10^{0}$ | $\mathbf{6.13 \times 10^{-1}}$ |

Table 2: Rollout Hausdorff distance

| System | Proposed | NeuralODE (Chen et al., 2018) | GRU-ODE-Bayes (De Brouwer et al., 2019) | CKF/RBSE Hybrid (Demirkaya et al., 2021) |
|---|---|---|---|---|
| Harmonic Oscillator | $\mathbf{2.28 \times 10^{-2}}$ | $5.18 \times 10^{0}$ | $1.01 \times 10^{0}$ | $5.70 \times 10^{-2}$ |
| Hodgkin–Huxley Neuron | $\mathbf{6.30 \times 10^{-1}}$ | $1.11 \times 10^{1}$ | $1.08 \times 10^{0}$ | $1.56 \times 10^{0}$ |
| Brusselator | $\mathbf{3.65 \times 10^{-1}}$ | $2.76 \times 10^{0}$ | $9.64 \times 10^{-1}$ | $2.20 \times 10^{0}$ |
| Yeast Glycolysis | $\mathbf{7.69 \times 10^{-3}}$ | $4.39 \times 10^{1}$ | $2.85 \times 10^{-1}$ | $5.59 \times 10^{-2}$ |
| Retinal Circulation | $\mathbf{1.88 \times 10^{0}}$ | $4.85 \times 10^{0}$ | $1.95 \times 10^{0}$ | $5.13 \times 10^{1}$ |

## 4.6 Robustness to Measurement Noise

**Testing across noise levels.** To evaluate robustness under increasingly noisy observations, we repeated the state-estimation and learning pipeline across a range of measurement noise levels, controlled via the signal-to-noise ratio (SNR) defined in Eq. 23. This experiment isolates a key benefit of using a Kalman-style smoother: even when the observations are heavily corrupted, the RTS smoother can still leverage the known dynamical structure and temporal coupling to infer a coherent latent trajectory.

**Recovering missing states under high noise.** Figure 10 demonstrates this effect on the Brusselator system. As the SNR decreases (i.e., noise increases), the observation sequences become progressively less informative. Nevertheless, the smoother continues to produce accurate reconstructions of the full state trajectory, including the *missing/unobserved* component, remaining close to the ground-truth dynamics even at high noise (e.g., SNR= 5). This illustrates the intended role of RTS guidance in our framework: providing dynamics-consistent pseudo-latent trajectories and initial conditions that remain reliable when encoder-based inference tends to become unstable. Beyond the Brusselator, we repeat the same SNR sweep on the higher-dimensional yeast glycolysis oscillator and on the stiff Hodgkin–Huxley neuron model, observing the same qualitative behavior: RTS-guided learning remains stable under high noise. These additional noise results are summarized in Appendix C.1.1.

$$\text{SNR} = \frac{\sqrt{\mathbb{E}[x^2(t)]}}{\sqrt{\mathbb{E}[\epsilon^2(t)]}} \tag{23}$$

where $x(t)$ is the (clean) signal, $\epsilon(t)$ is the additive measurement noise, and $\mathbb{E}[\cdot]$ denotes the time-average expectation over the simulated trajectory.

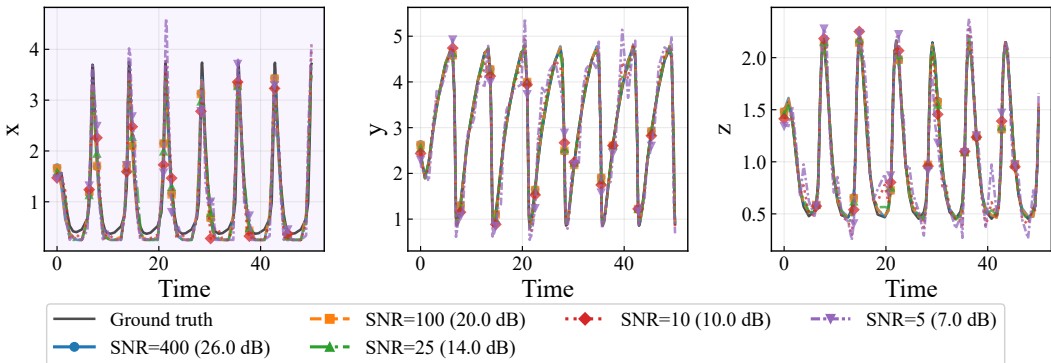

Figure 10: Brusselator: RTS state estimates across measurement noise levels. We vary the measurement noise according to different SNR values (higher SNR = lower noise). The black curve shows the ground-truth trajectory, while colored curves show RTS smoothed estimates obtained from noisy observations at each SNR. Despite increased noise at low SNR (e.g., SNR= 5), the smoother remains able to recover a coherent latent trajectory and accurately estimate the missing state, highlighting robustness of the Kalman-smoother-based inference in partially observed, noisy regimes.

## 4.7 Ablation Studies

To understand which components of the method are responsible for the performance gains, we ablate the three main parts of the training objective on two representative systems: the nonlinear Brusselator and the stiff Hodgkin–Huxley neuron. Starting from the full objective, we remove one component at a time: (i) *w/o latent-state loss*, where training uses only the observation-space rollout loss; (ii) *w/o covariance weighting*, where inverse-covariance weights are replaced by identity matrices; and (iii) *w/o RTS guidance*, where the backward smoother is removed and forward filtered estimates are used instead. Table 3 reports the resulting error on the unmeasured state together with a long-horizon rollout metric.

Table 3: Ablation of the main components of the proposed method. Lower is better. Latent RMSE is computed on the unmeasured state only; rollout NRMSE is computed over a 25-step forecast horizon in measurement space.

| Variant | Brusselator Latent RMSE ↓ | Hodgkin–Huxley Latent RMSE ↓ | 25-step Rollout NRMSE Avg. ↓ |
|---|---|---|---|
| Full objective | **0.233** | **0.052** | **0.118** |
| w/o latent-state loss | 0.301 | 0.071 | 0.154 |
| w/o covariance weighting | 0.286 | 0.064 | 0.141 |
| w/o RTS guidance | 0.412 | 0.109 | 0.221 |

The full model performs best across all settings. Removing the latent-state loss increases both latent reconstruction error and rollout error, showing that supervision in latent space is important under partial observability. Replacing covariance-based weights with uniform weights also hurts performance, indicating that uncertainty-aware weighting improves optimization stability. The largest drop occurs when RTS guidance is removed, confirming that smoothed trajectories provide more coherent pseudo-targets than forward filtered estimates alone. Overall, the ablation shows that all three components contribute, with RTS guidance having the strongest effect.

## 4.8 Summary of Empirical Findings

Across the benchmarks, the proposed method improved training stability in settings with partial measurements and reduced long-horizon rollout error relative to encoder-based baselines. The gains were most noticeable in systems where (i) measurements provide only indirect information about hidden states, and

(ii) small errors in latent-state estimates can lead to large downstream effects (e.g., stiff or strongly coupled dynamics). That said, the approach does not remove all failure modes: when measurements are extremely sparse, or when the assumed noise model is a poor fit, the resulting latent trajectories can be biased and this can carry into the learned dynamics.

# 5 Discussion and Limitations

## 5.1 Computational Cost and Scalability

Computationally, each RTS pass scales linearly with the total number of time points and subjects, and approximately cubically with state dimension due to covariance factorizations (roughly $\mathcal{O}(\sum_i T_i d^3)$), with a moderate constant from $2d$ cubature points. The neural rollout update adds cost proportional to the number of rollout horizons and ODE solver evaluations.

However, this additional cost is concentrated at training time. A practical advantage of the proposed framework is that the final model does not require filtering, smoothing, or a learned recognition network at inference time: after training, it is simply a hybrid ODE that can be forward simulated from an initial condition. This makes the method appealing in applications where heavier offline training is acceptable, but deployment should remain simple, interpretable, and computationally lightweight. Future work could further improve scalability through reduced-rank covariance approximations, windowed smoothing, or more efficient state-estimation schemes for higher-dimensional systems.

## 5.2 Modeling Assumptions

A key modeling assumption in this work is that, for each benchmark, we replace the dynamics of a single state with a neural network while keeping the remaining state equations mechanistic. In many ODE systems, more than one state might me unmeasured. In such cases, one can extend this work to estimate multiple unmeasured states simultaneously. Similar extensions have been proposed in Demirkaya et al. (2024), where they replace two states simultaneously.

## 5.3 Limitations

While the proposed method performs well across the benchmarks considered, several limitations remain. First, when measurements cover only a small fraction of the state space, or when the measurement function $\boldsymbol{h}$ provides only indirect or weak information about the latent states, the smoother may not have enough information to produce reliable latent trajectories, and the resulting pseudo-targets can mislead the parameter update. In all of our experiments the measurement function $\boldsymbol{h}$ is assumed known; settings where $\boldsymbol{h}$ is itself partially unknown or must be learned jointly with the dynamics are not addressed in this work. Second, the RTS smoother assumes Gaussian process and measurement noise; when the true noise is non-Gaussian or state-dependent, the smoothed estimates can be systematically biased, and this bias propagates into the learned dynamics. Third, the alternating optimization scheme lacks formal convergence guarantees: although we observe stable convergence across all experiments, the procedure may in principle oscillate or converge to a poor local minimum, particularly when the initial dynamics model $\boldsymbol{f}_{\theta^{(0)}}$ is far from the true system. Finally, our experiments replace the dynamics of a single state variable at a time; extending to systems where multiple state equations are simultaneously unknown introduces additional identifiability challenges. As mentioned in sec. 5.2, prior work has shown that replacing two states simultaneously is feasible in specific physiological settings (Demirkaya et al., 2024), but the general case requires further investigation.

# 6 Conclusion

We introduced an RTS smoother-guided approach for learning hybrid neural–physics ODEs from partial and noisy observations, where only some components of the system state are measured. The key idea is to use RTS smoothing during training to estimate latent trajectories and uncertainties, and to use these estimates

to supervise the unknown part of the dynamics. At test time, however, the learned model is simply a standalone hybrid ODE that can be simulated directly, without requiring an online state estimator.

The main motivation for this setting is missing-state learning. In many scientific systems, only a subset of the state is observed, while the unmeasured variables are exactly the ones needed to identify the unknown dynamics. Across the benchmark systems, the proposed method gives strong performance on unmeasured-state reconstruction and is generally more reliable than standard latent neural ODE baselines under partial observability. The gains are especially clear on systems where the hidden variables are tightly coupled to the measured ones, suggesting that smoother-based latent targets provide a useful training signal for recovering missing state structure.

This also helps position our method relative to prior work. Unlike NeuralODE and GRU-ODE-Bayes, our approach does not depend on a learned inference network to represent latent trajectories during training (Chen et al., 2018; De Brouwer et al., 2019). And unlike estimator-focused methods such as KalmanNet, RTSNet, and related CKF-based hybrid approaches, our aim is not to learn a better filter or smoother, but to use smoothing as a training mechanism for learning deployable dynamics models (Revach et al., 2022; 2023; Demirkaya et al., 2021; 2024).

The empirical results support this view. Across benchmark systems, the proposed method accurately reconstructs missing states from partial observations, as illustrated by the state-estimation results in Figures 3 to 7. It also remains robust as measurement noise increases, with Figure 10 showing that the smoother continues to recover coherent latent trajectories even at low SNR. The ablation study in Table 3 further shows that each proposed component contributes to performance, with additional gains from latent-state supervision and covariance-aware weighting. Finally, the long-horizon rollout results in Figure 9 and the quantitative comparisons in Tables 1 and 2 show that the learned models remain closely aligned with the true dynamics across a variety of systems. Overall, these results suggest that classical smoothing can be a useful ingredient for training interpretable hybrid differential models when part of the system state is never observed.

Several directions remain open. The current framework assumes a known measurement function and Gaussian noise, and our experiments replace a single state equation at a time; relaxing these assumptions — for example, by jointly learning components of the measurement model, accommodating non-Gaussian noise, or replacing multiple state equations simultaneously — would broaden the applicability of the approach. Overall, the results suggest that classical smoothing can be a useful ingredient for training interpretable hybrid differential models when part of the system state is never observed.

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

## Appendix

## A    CKF and RTS Smoother Details

This appendix summarizes the state-estimation machinery underlying the smoother operator $\mathcal{S}_\phi$ used in Section 3.1. During training, we use a Cubature Kalman Filter (CKF) for forward state estimation and a Rauch–Tung–Striebel (RTS) smoother for backward refinement. The resulting smoothed states $\hat{\boldsymbol{x}}_k^{(i)}(\theta)$ and covariances $\mathbf{P}_k^{(i)}(\theta)$ are used as rollout initializers and to define the covariance-based weights in the training objective.

### A.1    Discretized nonlinear state-space model

We start from the continuous-time model

$$\dot{\boldsymbol{x}}(t) = f_\theta(\boldsymbol{x}(t)) + \boldsymbol{w}(t), \qquad \boldsymbol{y}(t_k) = h(\boldsymbol{x}(t_k)) + \boldsymbol{\varepsilon}_k,$$

with measurement-noise covariance $\mathbf{R}_k$ and (discretized) process-noise covariance $\mathbf{Q}_k$ over $\Delta t_k = t_{k+1} - t_k$.

**ODE flow map.**    Define the one-step ODE flow map over the sampling interval $\Delta t_k$ as

$$\Phi_\theta(\boldsymbol{x}_k, \Delta t_k) := \boldsymbol{x}_k + \int_{t_k}^{t_{k+1}} f_\theta(\boldsymbol{x}(t; \boldsymbol{x}_k, \theta)) \, dt, \qquad \boldsymbol{x}(t_k) = \boldsymbol{x}_k, \qquad \Delta t_k = t_{k+1} - t_k,$$

i.e., the state returned by an ODE solver at $t_{k+1}$ when integrating $\dot{\boldsymbol{x}} = f_\theta(\boldsymbol{x})$ from initial condition $\boldsymbol{x}_k$ at time $t_k$.

This yields the discrete-time nonlinear state-space model

$$\boldsymbol{x}_{k+1} = \Phi_\theta(\boldsymbol{x}_k, \Delta t_k) + \boldsymbol{w}_k, \qquad \boldsymbol{y}_k = h(\boldsymbol{x}_k) + \boldsymbol{\varepsilon}_k, \tag{24}$$

where $\boldsymbol{w}_k \sim \mathcal{N}(\boldsymbol{0}, \mathbf{Q}_k)$ and $\boldsymbol{\varepsilon}_k \sim \mathcal{N}(\boldsymbol{0}, \mathbf{R}_k)$.

## A.2 Cubature Kalman Filter (CKF)

Let $d = \dim(\boldsymbol{x})$. At time $k$, the filtered estimate is $\boldsymbol{x}_{k|k}(\theta)$ with covariance $\mathbf{P}_{k|k}(\theta)$. Let $\mathbf{P}_{k|k}(\theta) = \mathbf{S}_{k|k}\mathbf{S}_{k|k}^\top$ and use spherical–radial cubature points

$$\xi_j \in \{\pm\sqrt{d}\,\mathbf{e}_1, \dots, \pm\sqrt{d}\,\mathbf{e}_d\}, \qquad w_j = \frac{1}{2d}.$$

Form state cubature points

$$\mathbf{X}_{k|k}^{(j)} = \boldsymbol{x}_{k|k}(\theta) + \mathbf{S}_{k|k}\xi_j.$$

**Prediction.** Propagate through the transition (explicit $\Delta t_k$) and compute predicted moments:

$$\mathbf{X}_{k+1|k}^{(j)} = \Phi_\theta\left(\mathbf{X}_{k|k}^{(j)}, \Delta t_k\right), \qquad \boldsymbol{x}_{k+1|k}(\theta) = \sum_{j=1}^{2d} w_j\,\mathbf{X}_{k+1|k}^{(j)},$$

$$\mathbf{P}_{k+1|k}(\theta) = \sum_{j=1}^{2d} w_j\left(\mathbf{X}_{k+1|k}^{(j)} - \boldsymbol{x}_{k+1|k}(\theta)\right)\left(\mathbf{X}_{k+1|k}^{(j)} - \boldsymbol{x}_{k+1|k}(\theta)\right)^\top + \mathbf{Q}_k.$$

**Update.** Map predicted points through $h$:

$$\mathbf{Y}_{k+1|k}^{(j)} = h\left(\mathbf{X}_{k+1|k}^{(j)}\right), \qquad \boldsymbol{y}_{k+1|k}(\theta) = \sum_{j=1}^{2d} w_j\,\mathbf{Y}_{k+1|k}^{(j)}.$$

Compute innovation and cross-covariances:

$$\mathbf{P}_{yy,k+1} = \sum_{j=1}^{2d} w_j\left(\mathbf{Y}_{k+1|k}^{(j)} - \boldsymbol{y}_{k+1|k}(\theta)\right)\left(\mathbf{Y}_{k+1|k}^{(j)} - \boldsymbol{y}_{k+1|k}(\theta)\right)^\top + \mathbf{R}_{k+1},$$

$$\mathbf{P}_{xy,k+1} = \sum_{j=1}^{2d} w_j\left(\mathbf{X}_{k+1|k}^{(j)} - \boldsymbol{x}_{k+1|k}(\theta)\right)\left(\mathbf{Y}_{k+1|k}^{(j)} - \boldsymbol{y}_{k+1|k}(\theta)\right)^\top.$$

Then

$$\mathbf{K}_{k+1} = \mathbf{P}_{xy,k+1}\mathbf{P}_{yy,k+1}^{-1}, \qquad \boldsymbol{x}_{k+1|k+1}(\theta) = \boldsymbol{x}_{k+1|k}(\theta) + \mathbf{K}_{k+1}\left(\boldsymbol{y}_{k+1} - \boldsymbol{y}_{k+1|k}(\theta)\right),$$

$$\mathbf{P}_{k+1|k+1}(\theta) = \mathbf{P}_{k+1|k}(\theta) - \mathbf{K}_{k+1}\mathbf{P}_{yy,k+1}\mathbf{K}_{k+1}^\top.$$

## A.3 Rauch–Tung–Striebel smoothing (CKF–RTS)

After the CKF forward pass, the RTS backward recursion produces smoothed means and covariances. Initialize

$$\hat{\boldsymbol{x}}_{T_i}(\theta) = \boldsymbol{x}_{T_i|T_i}(\theta), \qquad \mathbf{P}_{T_i}(\theta) = \mathbf{P}_{T_i|T_i}(\theta).$$

For $k = T_i - 1, \dots, 0$, compute the state cross-covariance

$$\mathbf{P}_{k,k+1|k}(\theta) = \sum_{j=1}^{2d} w_j\left(\mathbf{X}_{k|k}^{(j)} - \boldsymbol{x}_{k|k}(\theta)\right)\left(\mathbf{X}_{k+1|k}^{(j)} - \boldsymbol{x}_{k+1|k}(\theta)\right)^\top,$$

where

$$\mathbf{X}_{k+1|k}^{(j)} = \Phi_\theta\left(\mathbf{X}_{k|k}^{(j)}, \Delta t_k\right).$$

The smoother gain is

$$\mathbf{G}_k(\theta) = \mathbf{P}_{k,k+1|k}(\theta)\,\mathbf{P}_{k+1|k}(\theta)^{-1}.$$

The smoothed mean and covariance are

$$\hat{\boldsymbol{x}}_k(\theta) = \boldsymbol{x}_{k|k}(\theta) + \mathbf{G}_k(\theta)\big(\hat{\boldsymbol{x}}_{k+1}(\theta) - \boldsymbol{x}_{k+1|k}(\theta)\big),$$

$$\mathbf{P}_k(\theta) = \mathbf{P}_{k|k}(\theta) + \mathbf{G}_k(\theta)\big(\mathbf{P}_{k+1}(\theta) - \mathbf{P}_{k+1|k}(\theta)\big)\mathbf{G}_k(\theta)^\top.$$

We denote the resulting smoother operator by

$$\hat{\boldsymbol{x}}_{0:T_i}(\theta),\ \mathbf{P}_{0:T_i}(\theta) = \mathcal{S}_\phi\big(\boldsymbol{y}_{1:T_i}; f_\theta, h\big),$$

which is used in the main text to initialize rollouts and to form the covariance-weighted objective.

### A.4   Initial-state interpretation and use in training

Under the Gaussian state-space model, the smoothed trajectory $\hat{\boldsymbol{x}}_{0:T_i}^{(i)}(\theta)$ can be interpreted as (approximately) maximizing the posterior density over trajectories:

$$\hat{\boldsymbol{x}}_{0:T_i}^{(i)}(\theta) \approx \underset{\boldsymbol{x}_{0:T_i}^{(i)}}{\arg\max}\ q_\phi\left(\boldsymbol{x}_{0:T_i}^{(i)} \mid \boldsymbol{y}_{1:T_i}^{(i)}; f_\theta, h\right),$$

where $q_\phi$ denotes the RTS posterior approximation induced by $\phi$. In particular, the smoothed initial state

$$\hat{\boldsymbol{x}}_0^{(i)}(\theta) = \underset{\boldsymbol{x}_0}{\arg\max}\ q_\phi\left(\boldsymbol{x}_0 \mid \boldsymbol{y}_{1:T_i}^{(i)}; f_\theta, h\right)$$

is simply the first component of $\hat{\boldsymbol{x}}_{0:T_i}^{(i)}(\theta)$; it is *not* an independently optimized variable.

In the main training procedure, these smoothed states and covariances serve three purposes:

1. **Rollout initialization:** we start ODE rollouts from $\hat{\boldsymbol{x}}_k^{(i)}(\theta)$ when computing multi-step predictions $\tilde{\boldsymbol{x}}_{k+n}^{(i,n)}(\theta)$ and $\tilde{\boldsymbol{y}}_{k+n}^{(i,n)}(\theta)$.

2. **Latent-state targets in the loss:** the smoothed states $\hat{\boldsymbol{x}}_{k+n}^{(i)}(\theta)$ act as structured pseudo-targets in the latent-state error term $\boldsymbol{e}_{k+n}^{(x,i,n)}(\theta) = \tilde{\boldsymbol{x}}_{k+n}^{(i,n)}(\theta) - \hat{\boldsymbol{x}}_{k+n}^{(i)}(\theta)$, encouraging the learned dynamics to reproduce the smoother trajectories.

3. **Covariance-based weighting:** the smoothed covariances $\mathbf{P}_k^{(i)}(\theta)$ define the precision weights $\mathbf{W}_{k+n}^{(x,i,n)}(\theta) = \big(\mathbf{P}_{k+n}^{(i)}(\theta)\big)^{-1}$ in the covariance-weighted least-squares objective, so that uncertain time steps contribute less to the latent-state loss.

Together, proposed method provides a structured, dynamics-aware mechanism for constraining latent trajectories, defining uncertainty-aware weights, and supplying pseudo-targets for the latent-state loss. The dynamics parameters $\theta$ are then learned by minimizing the rollout-based objective defined in the main text.

## B   ODE Systems and Hybrid Parameterizations

This appendix summarizes the ordinary differential equations (ODEs) used in our experiments and specifies which state components are replaced by neural networks in the hybrid physics–neural models described in Section 3.5. For each system we denote the latent state by $\boldsymbol{s}(t)$ and identify the subset of state derivatives that are parameterized by $f_{\theta,\mathrm{unk}}$.

### B.1  Harmonic Oscillator

We use a standard undamped harmonic oscillator with state $\boldsymbol{s}(t) = [q(t), v(t)]^\top$, where $q$ is position and $v$ is velocity:

$$\dot{q}(t) = v(t), \tag{25}$$

$$\dot{v}(t) = -\omega_0^2\, q(t), \tag{26}$$

with natural frequency $\omega_0 > 0$. We measure only the position,

$$y(t) = h(\boldsymbol{s}(t)) = q(t). \tag{27}$$

In the hybrid model we keep the kinematic relation $\dot{q}(t) = v(t)$ and replace the acceleration by a neural network:

$$\dot{q}(t) = v(t), \tag{28}$$

$$\dot{v}(t) = f_{\theta,\mathrm{unk}}\big(q(t), v(t)\big). \tag{29}$$

**Related ML usage and rationale**  Harmonic oscillators are a standard benchmark in the Neural ODE literature for testing integration, stability, and latent-state recovery (Zhu et al., 2022). They provide the simplest partially observed setting in this paper: a two-dimensional system with one measured state and one latent state. In our hybrid formulation, we preserve the exact relation $\dot{q} = v$ and learn only the acceleration term $\dot{v}$. This makes the example a clean test of whether the proposed method can recover latent velocity from noisy position measurements and reproduce the expected oscillator dynamics.

### B.2  Hodgkin–Huxley Neuron

We use the classical Hodgkin–Huxley model with state $\boldsymbol{s}(t) = [V(t), m(t), h(t), n(t)]^\top$, where $V$ is membrane voltage and $m, h, n$ are gating variables (Hodgkin & Huxley, 1952). The ODEs are

$$C_m \frac{dV}{dt} = I_{\mathrm{ext}}(t) - g_{\mathrm{Na}} m^3 h\left(V - E_{\mathrm{Na}}\right) - g_{\mathrm{K}} n^4 \left(V - E_{\mathrm{K}}\right) - g_{\mathrm{L}}\left(V - E_{\mathrm{L}}\right), \tag{30}$$

$$\frac{dm}{dt} = \alpha_m(V)\left(1 - m\right) - \beta_m(V)\, m, \tag{31}$$

$$\frac{dh}{dt} = \alpha_h(V)\left(1 - h\right) - \beta_h(V)\, h, \tag{32}$$

$$\frac{dn}{dt} = \alpha_n(V)\left(1 - n\right) - \beta_n(V)\, n. \tag{33}$$

We measure all states except $m$:

$$y(t) = h(\boldsymbol{s}(t)) = \begin{bmatrix} V(t) \\ h(t) \\ n(t) \end{bmatrix}. \tag{34}$$

**Hybrid replacement.**  In the proposed hybrid model we keep the voltage equation and the $(h, n)$ gating dynamics explicit, and replace only the *second state* dynamics (the $m$-gate):

$$C_m \frac{dV}{dt} = I_{\mathrm{ext}}(t) - g_{\mathrm{Na}} m^3 h\left(V - E_{\mathrm{Na}}\right) - g_{\mathrm{K}} n^4 \left(V - E_{\mathrm{K}}\right) - g_{\mathrm{L}}\left(V - E_{\mathrm{L}}\right), \tag{35}$$

$$\frac{dm}{dt} = f_{\theta,\mathrm{unk}}\big(V(t), m(t), h(t), n(t)\big), \tag{36}$$

$$\frac{dh}{dt} = \alpha_h(V)\left(1 - h\right) - \beta_h(V)\, h, \tag{37}$$

$$\frac{dn}{dt} = \alpha_n(V)\left(1 - n\right) - \beta_n(V)\, n. \tag{38}$$

**Related ML usage and rationale.** The Hodgkin–Huxley system is a common benchmark for neural differential equation methods because it combines stiff dynamics, strong nonlinear coupling, and partially observed biophysical states (e.g., (Ghanem et al., 2025; Demirkaya et al., 2024; Centofanti et al., 2024; Tanoh et al., 2025; Lei & Mirams, 2021)). In our setup, $V$, $h$, and $n$ are observed, while the sodium activation gate $m$ remains latent. We replace only the $m$-gate dynamics because $m$ controls fast sodium activation and plays a central role in spike initiation. This makes the benchmark a focused test of whether the proposed method can recover a single but highly influential latent state while preserving the remaining Hodgkin–Huxley structure.

### B.3 Retinal Circulation

For retinal hemodynamics we use a reduced lumped circuit with four pressure states,

$$\boldsymbol{s}(t) = [P_1(t), P_2(t), P_4(t), P_5(t)]^\top, \tag{39}$$

representing pressures in proximal arterial, distal arterial, capillary, and venous compartments. A simplified form of the ODEs is

$$C_1 \frac{dP_1}{dt} = \frac{P_{\text{in}}(t) - P_1}{R_{\text{in}}} - \frac{P_1 - P_2}{R_1(P_1)}, \tag{40}$$

$$C_2 \frac{dP_2}{dt} = \frac{P_1 - P_2}{R_1(P_1)} - \frac{P_2 - P_4}{R_2(P_2)}, \tag{41}$$

$$C_4 \frac{dP_4}{dt} = \frac{P_2 - P_4}{R_2(P_2)} - \frac{P_4 - P_5}{R_4(P_4)}, \tag{42}$$

$$C_5 \frac{dP_5}{dt} = \frac{P_4 - P_5}{R_4(P_4)} - \frac{P_5 - P_{\text{out}}}{R_{\text{out}}}, \tag{43}$$

where $C_i$ are compliances, $R_{\text{in}}$ and $R_{\text{out}}$ are fixed inlet/outlet resistances, and $R_1, R_2, R_4$ are pressure-dependent microvascular resistances modelling autoregulation and vessel collapsibility. The measurement vector collects a subset of pressures or flows, e.g.,

$$y(t) = h(\boldsymbol{s}(t)) = \begin{bmatrix} P_1(t) \\ Q_{\text{ret}}(t) \end{bmatrix}, \qquad Q_{\text{ret}}(t) = \frac{P_1(t) - P_{\text{out}}}{R_{\text{eq}}(P_1, P_2, P_4, P_5)}. \tag{44}$$

In the hybrid model we keep the proximal and distal arterial dynamics and the venous outflow explicit, and replace the capillary compartment ODE by a neural network:

$$C_1 \frac{dP_1}{dt} = \frac{P_{\text{in}}(t) - P_1}{R_{\text{in}}} - \frac{P_1 - P_2}{R_1(P_1)}, \tag{45}$$

$$C_2 \frac{dP_2}{dt} = \frac{P_1 - P_2}{R_1(P_1)} - \frac{P_2 - P_4}{R_2(P_2)}, \tag{46}$$

$$\frac{dP_4}{dt} = f_{\theta,\text{unk}}\big(P_1(t), P_2(t), P_4(t), P_5(t), P_{\text{in}}(t)\big), \tag{47}$$

$$C_5 \frac{dP_5}{dt} = \frac{P_4 - P_5}{R_4(P_4)} - \frac{P_5 - P_{\text{out}}}{R_{\text{out}}}. \tag{48}$$

**Related ML usage and rationale** The retinal circulation model gives us a nonlinear, moderately sized state-space system with partial observations and explicit mechanistic structure, which is useful for testing hybrid ODE methods beyond simple toy setups. Similar retinal models have already been used as case studies for CKF-based hybrid ODE–NN approaches (Demirkaya et al., 2021; 2024), so it is a natural benchmark for the proposed RTS-guided variant. In our configuration the capillary compartment $P_4$ is not directly observed and is the main source of model mismatch and inter-subject variability, while the upstream and downstream equations are relatively well specified. We therefore keep the proximal/distal arterial and venous equations explicit and replace only $dP_4/dt$ with a neural term, concentrating learning where the physics is least reliable and using RTS to regularize the latent state through the remaining analytical structure.

### B.4 Brusselator Reaction Model

For the Brusselator we consider a three-species extension of the classical autocatalytic reaction network with state $\boldsymbol{s}(t) = [u_1(t), u_2(t), u_3(t)]^\top$:

$$\frac{du_1}{dt} = A - (B+1)\,u_1 + u_1^2 u_2, \tag{49}$$

$$\frac{du_2}{dt} = B\,u_1 - u_1^2 u_2 - u_2 + k\,u_3, \tag{50}$$

$$\frac{du_3}{dt} = -k\,u_3 + \gamma\,u_2, \tag{51}$$

with feed parameters $A, B$ and coupling parameters $k, \gamma$. We assume measurements

$$y(t) = h(\boldsymbol{s}(t)) = \begin{bmatrix} u_2(t) \\ u_3(t) \end{bmatrix}. \tag{52}$$

In the hybrid neural differential model we keep the $u_2$ and $u_3$ equations explicit and replace the $u_1$ dynamics:

$$\frac{du_1}{dt} = f_{\theta,\mathrm{unk}}\big(u_1(t), u_2(t), u_3(t)\big), \tag{53}$$

$$\frac{du_2}{dt} = B\,u_1 - u_1^2 u_2 - u_2 + k\,u_3, \tag{54}$$

$$\frac{du_3}{dt} = -k\,u_3 + \gamma\,u_2. \tag{55}$$

**Brusselator hybrid replacement rationale.** For the Brusselator, we retain the known mechanistic equations for the observed species and replace the differential equation of the unmeasured species with a neural component. Concretely, if the state is written as $\boldsymbol{x}(t) = [x_1(t), x_2(t)]^\top$ and only one component is measured, we model the hybrid dynamics as

$$\dot{x}_{\mathrm{obs}} = f_{\mathrm{phys}}\big(x_{\mathrm{obs}}, x_{\mathrm{lat}}\big), \qquad \dot{x}_{\mathrm{lat}} = f_{\mathrm{unk}}\big(x_{\mathrm{obs}}, x_{\mathrm{lat}}; \theta\big),$$

where $f_{\mathrm{phys}}(\cdot)$ denotes the retained mechanistic reaction term and $f_{\mathrm{unk}}(\cdot; \theta)$ is a neural network used to represent the missing dynamics of the latent species. This replacement is well motivated in the Brusselator because the system is low dimensional, strongly coupled, and exhibits a characteristic oscillatory limit cycle. As a result, errors in the latent-state dynamics directly distort the geometry of the trajectory in phase space, making the benchmark a clean test of whether the learned hybrid model can infer the missing state equation from partial observations while preserving the qualitative structure of the underlying nonlinear dynamics.

### B.5 Yeast Glycolysis Model

We consider a seven-dimensional yeast glycolysis oscillator with state $\boldsymbol{s}(t) = [x_1(t), x_2(t), x_3(t), x_4(t), x_5(t), x_6(t), x_7(t)]^\top$.

**Dynamics.** With fixed constants $\{c_i, d_i, e_i, f_i, g_i, h_i, j_i\}$ (given in the main text/config), the ODE is

$$\dot{x}_1 = c_1 + \frac{c_2\,x_1 x_6}{1 + c_3 x_6^4}, \tag{56}$$

$$\dot{x}_2 = \frac{d_1\,x_1 x_6}{1 + d_2 x_6^4} + d_3 x_2 - d_4 x_2 x_7, \tag{57}$$

$$\dot{x}_3 = e_1 x_2 + e_2 x_3 + e_3 x_2 x_7 + e_4 x_3 x_6, \tag{58}$$

$$\dot{x}_4 = f_1 x_3 + f_2 x_4 + f_3 x_5 + f_4 x_3 x_6 + f_5 x_4 x_7, \tag{59}$$

$$\dot{x}_5 = g_1 x_4 + g_2 x_5, \tag{60}$$

$$\dot{x}_6 = h_3 x_3 + h_5 x_6 + h_4 x_3 x_6 + \frac{h_1\,x_1 x_6}{1 + h_2 x_6^4}, \tag{61}$$

$$\dot{x}_7 = j_1 x_2 + j_2 x_2 x_7 + j_3 x_4 x_7. \tag{62}$$

**Measurement model.** All states except $x_4$ are measured:

$$y(t) = h(\boldsymbol{s}(t)) = [x_1(t), x_2(t), x_3(t), x_5(t), x_6(t), x_7(t)]^\top.$$

## C Additional Experiments and Results

### C.1 Additional Experiments

#### C.1.1 Robustness to measurement noise

This appendix complements Section 4.6 with an additional SNR sweep on the **Hodgkin–Huxley neuron**. Consistent with the main results, the proposed method recovers coherent latent trajectories even at low SNR, which helps keep the learned hybrid ODE rollouts stable under heavy measurement noise.

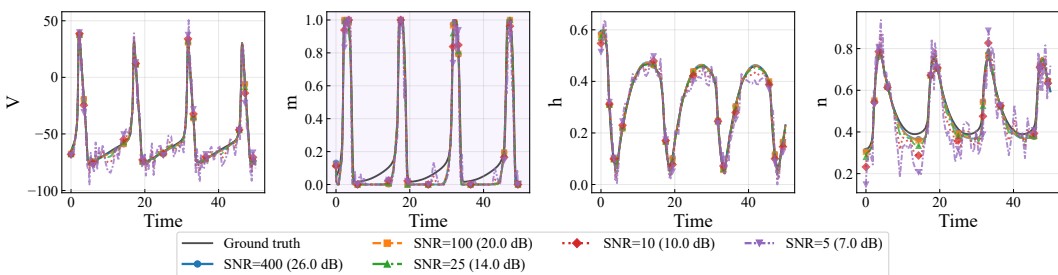

Figure 11: Hodgkin–Huxley neuron: RTS smoothed state estimates across measurement noise levels (SNR). Black curves show ground truth; colored curves show smoothed estimates from noisy observations. Even at low SNR, RTS produces stable latent trajectories that support robust hybrid-model learning.

### C.2 Software

We make the code public, and provide instructions on how to embed your physics equations into the code. The repository also provides instructions to how to load your data and use it with your custom models.

