# OpenReview forum: "RTS Smoother-Guided Learning of Physics-Based Neural Differential Models"
_TMLR — Under review for TMLR_

### Review · Reviewer_9QVp · 2026-05-29

**Summary Of Contributions:**

The paper proposes a two-stage method to learn the dynamics of an ODE system from partial and noisy observations when some state variables are hidden and parameters are unknown. In the hybrid neural–physics model, the known ODE terms are kept explicit, while the missing component of the dynamics is represented by a neural network. Its main contribution is an RTS smoother-guided training procedure. The method alternates between two steps: first, it uses a CKF/RTS smoother to estimate hidden state trajectories and their uncertainty; second, it treats those smoothed trajectories as pseudo-targets and updates the neural dynamics parameters by backpropagation. Also, the model is trained not only to match one-step dynamics, but also to match smoothed latent states and observations at multiple rollout horizons. The latent-state error is weighted by the inverse RTS smoother covariance, so uncertain pseudo-targets contribute less strongly to the loss. Empirically, the paper evaluates on five systems: harmonic oscillator, Hodgkin–Huxley, retinal circulation, Brusselator, and yeast glycolysis. These cover linear, nonlinear, oscillatory, physiological, higher-dimensional, and stiff dynamics. The proposed method generally outperforms benchmarks on hidden-state RMSE and rollout Hausdorff distance, with ablations suggesting that RTS guidance, latent-state loss, and covariance weighting all help.

### Strength:

- Interpretability. The method learns only the missing state equation and keeps the known ODE terms explicit.
- Multi-horizon learning. The multi-horizon objective makes the method more practical in terms of long-horizon prediction
- 5 test cases covering wide scenarios
- No need for online smoothing.

### Weaknesses

- Miss a few strong baseline and related study discussion.
- All tests are simulations, the application in real data is unknown.

**Audience:**

Yes

**Audience Explanation:**

The problem is important and well motivated. The interpretability of the method also makes it meaningful and practically relevant.

**Claims And Evidence:**

No

**Claims Explanation:**

Mainly because of a few missing points in the baselines and experiment designs. See below.

**Requested Changes:**

- Recognition ODE baseline is claimed but not included in the result. Please add.
- All experiments only have one missing state variables, add experiments with multiple missing state equations.
- Add comparison to functional-prior work and other PINN related work from Dr. Karniadakis, as a baseline or in the discussion section.
- Clarify how initial states are handled at train and test time.
- Could you explain a bit more what do you mean in Figure 9: "Each panel shows the best selected prediction over the displayed time window."

---

### Review · Reviewer_QxkR · 2026-06-11

**Summary Of Contributions:**

The paper proposes an RTS smoother-guided framework for learning hybrid neural-physics ODE models from partial and noisy observations. The method alternates between estimating latent state trajectories using an RTS smoother and updating the unknown neural component of the ODE using the smoothed trajectories as pseudo-targets. The final learned model preserves the known mechanistic dynamics while replacing missing or unknown state equations with neural network components.

Key strengths:

* Simple, natural, and interpretable idea that fits the hybrid neural-physics setting well.
* Strong empirical performance on most benchmarks, with good comparisons against recent baselines.
* Useful coverage of nonlinear and stiff dynamical systems.

Key weaknesses:

* The RTS smoother step and the relation between the continuous-time and discrete-time formulations could be explained more clearly.
* Most experiments appear to replace only one state equation, so a more severely partially observed setting would make the empirical evaluation stronger.

**Audience:**

Yes

**Audience Explanation:**

Yes, researchers interested in neural ODEs, hybrid scientific machine learning, state-space models, and learning dynamics from partial observations would likely find the proposed RTS-guided training strategy useful and relevant.

**Claims And Evidence:**

Yes

**Claims Explanation:**

Yes, the main claims are generally supported by convincing empirical evidence across multiple benchmark systems, although the presentation would be stronger if the RTS smoother step and the more challenging partial-observation settings were explained and evaluated more clearly.

**Requested Changes:**

Major changes:

- Clarify the connection between the continuous-time model in (1) and the discrete-time model in (7), especially whether $f_\theta$ denotes the continuous-time vector field, a numerical flow map, or a discretized transition function.
- Explain the role of the process noise term $w_k$ in (7) relative to the continuous-time noise in (1).
- Expand the description of the RTS smoother in the main text, especially around (11), so that $\hat{x}_{0:T_i}$, $P_{0:T_i}$, and their role in the training objective are clear.
- Discuss Figure 5, where the proposed method appears worse than baselines on the observed retinal pressures $P_1, P_2,$ and $P_5$, while performing competitively on the latent pressure $P_4$.
- Add, if possible, an experiment with more than one unmeasured or neural-replaced state equation, or a setting with a smaller observed subset.

Minor changes:

- Define $R$ and $Q$ earlier, preferably below (6).
- Use consistent terminology for “measurement error” and “measurement noise.”
- Make the horizon notation consistent, since both $n$ and $\mathcal{H}$ are used.
- Specify how the integration in (12) is computed in practice, including the numerical solver and relevant settings.
- In Figure 2, consider adding one or two intermediate horizon lengths between $H=1$ and $H=50$.
- Fix notation in (20) and (21), where $\tilde{\mathcal{X}}$ appears to be used inconsistently with $\hat{\mathcal{X}}$.
- Fix the formatting issue in Figure 3(a), where the legend and the text “Time” appear misaligned.
- Correct minor grammar issues, such as “allowing us to creating” and “more than one state might me unmeasured.”

---

### Review · Reviewer_2Bek · 2026-07-18

**Summary Of Contributions:**

The paper proposes to learn a dynamical system by doing jointly state smoothing and ODE fitting by alternative optimisation and multi-horizon loss.

**Audience:**

No

**Audience Explanation:**

I think this paper eventually might find an interested audience, but currently the interest is not yet clear or demonstrated.

The overall method is framed as novel, but to me seems quite conventional. Isn’t this at its core just a dual version of state estimation (eg. Dual EKF): we learn both states and forward function? This is a classic and well studied problem in classic filtering and in more recent ML literature. The alternative optimisation is a simple trick that is well known, and traces itself back to expectation maximization. The multi-horizon losses also have known precedents, and partial or semi-mechanistic forwards as well. I think the paper is framing itself a novel ML method, but as a filtering method the novelties largely dissolve. The paper should be positioned more clearly in both the filtering and neural ODE domains to make the main novelties explicit. In its current form its not yet clear why this is interesting for either audience, or what really is the novelty here. This is excarberated by doing insufficient comparisons to both communities (only one filtering competitor, and mostly old and simple neural ODE competitors).

**Broader Impact Concerns:**

No issues.

**Claims And Evidence:**

No

**Claims Explanation:**

The claims made are not really clear, and are coming in two places in sections 1 and 2. The precise claims should be made explicit in one place.

The empirical claims are somewhat vague. Five systems are chosen, but not clear why this set of five systems or what research questions we are answering. The results indicate RBSE is often visually just as good as the proposed method, even if the tables show big differences. There needs to be much wider comparisons to both filtering methods and to neural ODE methods (especially ones that target semi-mechanistic or partially observed systems). The paper mentions many related works that could be compared against. The competing methods should be a mixture of both classic and recent methods (now only filtering competitor is from 2021).

**Requested Changes:**

The presentation needs improvements. The density notation is over only (x,y), leaving the rest of the variables implicit. It would be clearer to show all major variables in the notation. In eq 2 and 9 the likelihood terms are constant wrt theta, and play no role. Eq 10 couples the likelihood to the theta in a way that is clearly useful, but deviates from eq 9 without explanation. The density notation does not make the dual optimistaion of xhat and theta clear. Sec 3.3. moves to continuous integration, while sec 3.2. was using discrete time stepping. The notation of x(t) in eq 12 is wrong: x(t) is the true latent that is unknown.

In retinal experiment the model is not fitting the data, which indicates that something is perhaps wrong in the implementation, optimisation, or maybe even in the modelling assumptions.